# Fast Eating Is Associated with Increased BMI among High-School Students

**DOI:** 10.3390/nu13030880

**Published:** 2021-03-09

**Authors:** Petter Fagerberg, Evangelia Charmandari, Christos Diou, Rachel Heimeier, Youla Karavidopoulou, Penio Kassari, Evangelia Koukoula, Irini Lekka, Nicos Maglaveras, Christos Maramis, Ioannis Pagkalos, Vasileios Papapanagiotou, Katerina Riviou, Ioannis Sarafis, Athanasia Tragomalou, Ioannis Ioakimidis

**Affiliations:** 1Innovative Use of Mobile Phones to Promote Physical Activity and Nutrition across the Lifespan (The IMPACT) Research Group, Department of Biosciences and Nutrition, Karolinska Institutet, 14183 Huddinge, Sweden; Ioannis.Ioakimidis@ki.se; 2Division of Endocrinology, Metabolism and Diabetes, First Department of Pediatrics, National and Kapodistrian University of Athens Medical School, “Aghia Sophia” Children’s Hospital, 11527 Athens, Greece; evangelia.charmandari@googlemail.com (E.C.); peniokassari@gmail.com (P.K.); nansymou@hotmail.com (A.T.); 3Division of Endocrinology and Metabolism, Center of Clinical, Experimental Surgery and Translational Research, Biomedical Research Foundation of the Academy of Athens, 11527 Athens, Greece; 4Department of Informatics and Telematics, Harokopio University of Athens, 17778 Athens, Greece; cdiou@hua.gr; 5International English School, 11858 Stockholm, Sweden; rachel.heimeier.sodermalm@engelska.se; 6Lab of Medical Informatics, Aristotle University of Thessaloniki, 54124 Thessaloniki, Greece; youlak@auth.gr (Y.K.); lekka@med.auth.gr (I.L.); nicmag@auth.gr (N.M.); chmaramis@certh.gr (C.M.); 7Ekpaideftiria N. Mpakogianni, 41500 Larissa, Greece; koukoulaeva@gmail.com; 8Department of Nutritional Sciences and Dietetics, School of Health Sciences, International Hellenic University, 57400 Thessaloniki, Greece; ipagkalos@ihu.gr; 9Multimedia Understanding Group, Information Processing Laboratory, School of Electrical and Computer Engineering, Aristotle University of Thessaloniki, 54124 Thessaloniki, Greece; vassilis@mug.ee.auth.gr (V.P.); sarafis@mug.ee.auth.gr (I.S.); 10Ellinogermaniki Agogi, 15351 Pallini, Greece; kriviou@ea.gr

**Keywords:** eating rate, obesity, eating quickly, fast eating, eating speed, objective measures, self-reported, validation, high-school students, adolescents

## Abstract

Fast self-reported eating rate (SRER) has been associated with increased adiposity in children and adults. No studies have been conducted among high-school students, and SRER has not been validated vs. objective eating rate (OBER) in such populations. The objectives were to investigate (among high-school student populations) the association between OBER and BMI z-scores (BMIz), the validity of SRER vs. OBER, and potential differences in BMIz between SRER categories. Three studies were conducted. Study 1 included 116 Swedish students (mean ± SD age: 16.5 ± 0.8, 59% females) who were eating school lunch. Food intake and meal duration were objectively recorded, and OBER was calculated. Additionally, students provided SRER. Study 2 included students (*n* = 50, mean ± SD age: 16.7 ± 0.6, 58% females) from Study 1 who ate another objectively recorded school lunch. Study 3 included 1832 high-school students (mean ± SD age: 15.8 ± 0.9, 51% females) from Sweden (*n* = 748) and Greece (*n* = 1084) who provided SRER. In Study 1, students with BMIz ≥ 0 had faster OBER vs. students with BMIz < 0 (mean difference: +7.7 g/min or +27%, *p* = 0.012), while students with fast SRER had higher OBER vs. students with slow SRER (mean difference: +13.7 g/min or +56%, *p* = 0.001). However, there was “minimal” agreement between SRER and OBER categories (κ = 0.31, *p* < 0.001). In Study 2, OBER during lunch 1 had a “large” correlation with OBER during lunch 2 (*r* = 0.75, *p* < 0.001). In Study 3, fast SRER students had higher BMIz vs. slow SRER students (mean difference: 0.37, *p* < 0.001). Similar observations were found among both Swedish and Greek students. For the first time in high-school students, we confirm the association between fast eating and increased adiposity. Our validation analysis suggests that SRER could be used as a proxy for OBER in studies with large sample sizes on a group level. With smaller samples, OBER should be used instead. To assess eating rate on an individual level, OBER can be used while SRER should be avoided.

## 1. Introduction

Obesity is associated with severe health problems such as heart disease [1], diabetes mellitus type 2 [1], depression [2], and some cancers [3]. Together, they are among the leading causes of premature death in the world, while at the same time being preventable [4]. An estimated 2 billion people now suffer from obesity [5] and the associated economic costs have been compared to the cost of smoking, armed violence, war and terrorism combined, or approximately three percent of the global GDP [6]. The direct cause of obesity is overconsumption of energy from food in relation to the energy demands of the body [7]. However, the causes of overconsumption of energy are multifactorial [8,9].

Interestingly, various eating behaviors have been shown to increase energy intake and adiposity in humans [10,11,12]. For example, experimental studies suggest that a fast eating rate causes increased short-term food intake vs. a slow eating rate (random-effects standardized mean difference: 0.45 [13]). Moreover, objectively measured eating rate is a stronger explanatory variable for how much food high-school students eat during school lunch vs. subjective variables, such as changes in perceived fullness, as well as tastiness [14]. Objectively measured eating rate has also been shown to be associated with increased BMI and body fat among 4.5-year-old Singaporean children [15], as well as to moderate the association between important childhood obesity risk factors and adiposity outcomes at the age of 6 years [16].

A growing number of epidemiological studies have found a consistent association between self-reported fast eating rate and obesity (pooled odds ratio of 2.15 vs. slow eaters), as well as increased BMI (pooled mean difference in BMI = +1.78 kg/m2 vs. slow eaters [17]). Additionally, a randomized controlled trial in young people with obesity has shown benefits in training participants to eat slower and to take smaller food portions in treatment outcomes (i.e., 0.24 standard deviation score lower for both BMI and body fat vs. standard care after 18 months of follow-up [18]), suggesting that the association between faster eating rate and increased BMI might be causal. It should also be noted that disturbed eating speed is associated with eating disorders, either with excessively slow eating rates reported in anorexia nervosa or faster than normal eating rates in binge-eating disorder patients [19]. Furthermore, it has been proposed that long-term disturbance in eating behavior pattern is one of the causal factors for the development of eating disorders [20], with various clinical programs currently providing eating behavior training during treatment.

However, criticism has been raised against self-reported measures of dietary intake [21,22,23], and similar concerns exist regarding self-reported measures of eating rate. In the epidemiological literature related to eating rate and health outcomes, simple questionnaires asking people to estimate how fast they eat in comparison to “others” are used. Surprisingly, few studies have been conducted to investigate the concurrent validity of such questionnaires vs. objectively measured eating rate.

To date, three studies with relatively small sample sizes (*n* = ~60–80) have validated self-reported eating rate categories vs. objectively measured eating rate [24,25,26]. The results of these studies indicate that self-reported eating rate categories can differentiate groups with high vs. low objectively measured eating rate, while categorization on an individual level should be avoided due to “minimal” categorical agreement. All three of the above-mentioned studies were conducted in laboratory environments. No comparisons between objective and subjective eating rate have been performed in “real-life” situations, nor among high-school students. Therefore, the generalizability of these results can be questioned, and proper validation of self-reported eating rate is needed in such contexts.

Additionally, no studies have been conducted in Swedish or Greek populations. The epidemiological self-reported literature on eating rate is also location specific, as it has previously included predominantly Japanese populations (for example 18-year-old women [27], 29- to 39-month-old children [28] and middle-aged men and women [29]), as well as middle-aged women in New Zealand [30], Dutch adults [24], Singaporean adults [31] and South Korean adults [32]. Thus, data across additional regions and target populations are also needed to clarify the generalizability of past findings.

### Aims

The aims of our three studies (Study 1. Single school lunch study, Study 2. Repeated school lunch study, and Study 3. BigO cohort study) were:

Study 1. To investigate the association between objective eating rate, BMI z-scores, food intake and weight categories among high-school students, as well as to determine whether self-reported eating rate categories can distinguish groups of different objectively measured eating rates.

Study 2. To assess the concurrent validity of self-reported eating rate categories vs. objective eating rate categories, and to assess the test–retest reliability of both subjective and objective eating rates across repeated measures from the same high-school students.

Study 3. To distinguish differences in BMI z-scores among self-reported eating rate categories in larger populations of Swedish and Greek high-school students and to estimate its relation to BMI z-scores.

## 2. Materials and Methods

### 2.1. Study Design

A cross-sectional study design was used in studies 1 (the single school lunch study) and 3 (the BigO cohort study), while a repeated-measures, within-subject study design was used in Study 2 (the repeated school lunch study).

### 2.2. Setting

Studies 1 and 2 were conducted in the school lunch canteen environment at Internationella Engelska Gymnasieskolan Södermalm (IEGS) (Stockholm, Sweden). The studies were part of the multinational EU project SPLENDID, with the aim to develop systems for early detection and interventions for childhood obesity in school children [33,34]. IEGS is a high school located in the central area of Stockholm, Sweden. Recruitment took place during February 2015, December 2015, and April 2017 (see Figure 1 for the complete timeline for studies 1 to 3). Study 2 included a subset of Study 1 subjects coming back for a repeated meal in the same school environment during February and March 2016 (2–3 months after their initial meal). The third study used data from a separate multinational EU project BigO (Big data against childhood obesity, [35]). In the BigO project, a smartphone application was developed for school children and adolescents to gather data related to the development of childhood obesity. Data collection was supported through school-supported actions between March 2018 and June 2020. Presentations of the BigO project were conducted in selected schools in Sweden and in Greece, in collaboration with the local school personnel at each school.

### 2.3. Participants

In Study 1, students from six classes (187 students) at the IEGS high school were invited to participate in monitored school lunches. The outcome dataset with unique participants (non-repeated meals) included 15 students from 2015 (pilot meals), 97 students from December 2015 and early 2016 (this was the year when most of the data collection took place) and another 2 students from 2017 (students that did not participate in the preceding years). It is important to note that Study 1 only included unique participants, i.e., for students that participated in more than one year, only one meal (mainly the late 2015/early 2016 meal, secondly the early 2015 meal, and lastly the 2017 meal) was included in the dataset for Study 1.

In Study 2, students from four out of the six previously invited classes (in the late 2015 data collection) were invited to participate once again, with a subset of 50 students finally providing repeated meals two to three months later (February and March 2016).

For Study 3, the BigO project [35] supported the non-discriminative, large-scale, spontaneous recruitment of students (in this case aged 15–18 years) from different European countries, with the main volume of data collection taking place in Sweden and Greece. In Sweden, data collection included 748 students from two high schools, one in Stockholm (IEGS, *n* = 613) and one in Uppsala (NTI Gymnasiet, *n* = 135). In Greece (included students in total = 1084), data collection focused on three cities, Athens (Ellinogermaniki Agogi high school, *n* = 230), Larissa (Ekpaideutiria Mpakogianni, *n* = 111) and Thessaloniki (across 16 public and private high schools, *n* = 439). An additional 304 high-school students who participated in a multidisciplinary, personalized intervention program for the management of overweight and obesity (Out-patient Clinic for the Prevention and management of Overweight and Obesity in Childhood and Adolescence, First Department of Pediatrics “Aghia Sofia” Children’s’ Hospital, Athens, Greece) also contributed data.

In all cases, irrespective of the country where the data collection took place, local school administrators, teachers and clinicians handled the consent process and final study recruitment, with the remote support of the BigO researchers. All data collection in schools was supported through school-sponsored projects with subjects relevant to lifestyle monitoring, as well as to citizen science [36]. As a rule, the recruitment efforts in schools targeted the whole school population through school-wide project advertisements, but the specifics differed across schools due to local educational requirements and schedules. Out of these populations, 1909 of the consented students activated the BigO app in their personal smartphones (Android and iOS platforms were supported). Furthermore, 96% (1832) provided answers to the self-reported eating rate questionnaire that is analyzed in Study 3, as well as provided self-reported estimates about their weight and height that were the basis of their BMI z-score calculation (using 2007 World Health Organization reference charts).

All the presented studies aimed at “real-life”, all-inclusive student population analysis. Thus, recruitment took place in a non-discriminatory fashion, meaning that no inclusion/exclusion criteria existed more than being part of the included schools, be willing to take part in the study procedures, and providing informed consent.

### 2.4. Data Sources/Measurements

BMI z-scores calculations: During studies 1 and 2, study personnel weighed and measured the height (by use of a weight and height scales, Seca, Chino CA 91710, USA) of each student before taking part in the school lunch measurements. These measurements, together with students age and sex, enabled calculation of BMI z-scores. The BMI z-scores were derived from an online calculator (https://apps.cpeg-gcep.net/who2007_cpeg/ [37], accessed on 8 March 2021) that used the WHO reference charts for child growth [38]. Students were later categorized in two groups: (a) students with BMI z-scores < 0, and (b) students with a BMI z-score ≥ 0, in order to compare the results to those obtained in a previous study among children that used a similar categorization scheme [15].

Meal procedure at school: Upon arrival to the school lunch canteen environment, students were equipped with a portable food scale (Mandometer version 4 during early 2015 and Mandometer version 5 in the data collection that took place in the later part of 2015, early 2016 and 2017 [39]), a mobile phone, as well as a questionnaire in paper format. The food scale was used to record food mass intake in grams (accounting for food additions and leftovers). In the lunchroom environment, digital cameras (GoPRO) were placed in each corner of the room to assess each student’s meal duration. The food mass intake (g) was later divided by the meal duration in minutes (established from the video recordings) to calculate the objective eating rate (g/minute) of each individual student. The questionnaires were used to assess the subjective eating rate of each student. The groups of slow, medium, and fast self-reported eating rate categories were used in accordance with previous literature (see the paragraph below for specific questions used as well as the available answers) [17].

The BigO data: During Study 3, each participant was asked to provide an estimation of their current weight and height during the initial registration in the BigO app. Afterwards, they were requested to fill in an eating rate questionnaire item: “How fast do you eat in comparison to others?”. The user could then select one of five options: “Much slower than others”, “Slower than others”, “similar to others”, “Faster than others” or “Much faster than others”. Students who self-reported eating “Much slower than others” and “Slower than others” were later merged into the eating rate category “Slow” and those reporting eating “Faster than others” or “Much faster than others” were categorized as “Fast” in a similar fashion as in previous literature [17], while students reporting eating similar to others were labeled as “intermediate”.

### 2.5. Served Food

During studies 1 and 2, students were offered a standardized buffet lunch meal in their natural school canteen environment that included potatoes, beef patties, celery patties, fish (pollock), cream sauce, vegetables (such as sliced carrots, cucumber, lettuce, sprouts, olives), crisp bread, cottage cheese and jam, all ad libitum. Water and milk were also available ad libitum during the lunch. This type of buffet meal is typical for Swedish schools and is served in IEGS daily. The study setup has previously been described in greater detail; see [14,40].

### 2.6. Study Size

Since both EU projects (SPLENDID and BigO) were novel health technology projects, the sample sizes in studies 1–3 were determined by the available students during each round of recruitment for the development of the technology. Therefore, post hoc power analyses were conducted with different aims and main outcomes in mind. In Study 1, 100% power was achieved for the primary ANOVA outcome analysis of group level differences in objective eating rate between the three groups of self-reported eating rate (effect size *f* = 0.742, alfa error probability = 0.05, total sample size = 114, and three groups). In Study 3, 98.7% power was achieved in the primary ANOVA outcome analysis of differences in BMI z-scores among the self-reported eating rate categories (effect size *f* = 0.105, alfa error probability = 0.05, sample size = 1832, and three groups). G*Power 3.1.9.7 was used for post hoc power analyses [41].

### 2.7. Statistics

For Study 1, Pearson’s correlation was used to assess the association between objective eating rate and objective food mass intake. One-way analysis of variance (ANOVA) was used for both objective and subjective eating rate category differences (slow, intermediate, and fast) in objective food mass intake. Bonferroni post hoc test was conducted to assess specific group level differences. Independent sample t-test was used to assess group level differences in objective eating rate among the two weight categories (BMI z-scores < 0 vs. BMI z-scores ≥ 0).

Cohen’s weighted kappa analysis was used for categorical eating rate agreement [42], i.e., for agreement between self-reported vs. objective eating rate categories in Study 1, for agreement between self-reported eating rate categories during lunch 1 vs. self-reported eating rate categories during lunch 2 (in Study 2) and for objective eating rate categories during lunch 1 vs. objective eating rate categories during lunch 2 (in Study 2). Furthermore, Pearson’s correlation was used for test–retest reliability analysis in Study 2 (eating rate during repeated school lunch meals among the same subjects), while a publicly available spreadsheet was used for the calculation of other test–retest reliability measures (i.e., the systematic change in mean and the typical error of measurement) [43,44].

In Study 3 (BigO cohort), ANOVA was used for all statistical tests between the different groups of self-reported eating rate categories (slow, intermediate, and fast), with Bonferroni post hoc tests conducted to assess specific group level differences when the overall ANOVA model was significant.

The variance explained in the dependent variable was examined by partial eta squared statistics in all ANOVA analyses. For correlation analyses, a correlation coefficient of 0.0–0.1 was interpreted as “trivial”, 0.1–0.3 “small”, 0.3–0.5 “moderate”, 0.5–0.7 “large”, 0.7–0.9 “very large”, and 0.9–1 as “nearly perfect” [45]. For categorical agreement analyses, a Cohen’s weighted Kappa value of 0–0.20 was interpreted as “No agreement”, 0.21–0.39 “Minimal agreement”, 0.40–0.59 “Weak agreement”, 0.60–0.79 “Moderate agreement”, 0.80–0.90 “Strong agreement”, and above 0.9–1 as “Almost perfect agreement” [46].

IBM SPSS statistical software version 27 was used for all statistical tests in studies 1–3 and *p* < 0.05 was used as the threshold for statistical significance.

### 2.8. Bias

BMI z-scores were used in all analyses instead of BMI to consider the natural growth occurring among adolescents (i.e., BMI z-scores takes into account age and sex in addition to weight and height [47]). In Study 3, in addition to the total population-level analysis, students were also split into separate groups based on their country (Sweden and Greece) to assess potential differences in self-reported eating rate within each country. A similar procedure was also done for the students who were participating in a personalized multidisciplinary management of obesity in Athens, Greece (*n* = 304, mean BMI Z-score: 2.19, standard deviation: 0.89).

## 3. Results

### 3.1. Participants

For specifics on number of students at each stage of studies 1–3, see Figure 2 below.

### 3.2. Descriptive Data

For descriptive statistics related to the included students in studies 1–3, see Table 1.

### 3.3. Main Results

#### 3.3.1. Study 1. Single School Lunch Study

##### Association between Objective Eating Rate and Objective Food Mass Intake (g)

There was a significant “large” correlation (Pearson’s *r* = 0.667, *p* < 0.001) between eating rate (g/min) and food mass intake (g) during the school lunch (see Figure 3A).

##### Association between Objective Eating Rate (g/min) and BMI z-Scores

There was also a significant “moderate” correlation (Pearson’s *r* = 0.310, *p* = 0.001) between objective eating rate (g/min) and BMI z-scores during the school lunch (see Figure 3B).

##### Objective Food Mass Intake among Objectively Established Eating Rate Categories

There was a significant difference in objectively measured food mass intake (g) between the objectively established eating rate categories [F(2, 111) = 30.578, *p* < 0.001, partial η^2^ = 0.355]. Post hoc comparisons using Bonferroni revealed that there was a significant difference between slow and intermediate [mean difference = −133 g, 95% CI = −210 g to −56 g; *p* < 0.001)] slow and fast [mean difference = −247 g, 95% CI = −324 g to −170 g; *p* < 0.001)] and between intermediate and fast [mean difference = −114 g, 95% CI = −191 g to −37 g; *p* = 0.01)] (Figure 4).

##### Objective Eating Rate among BMI z-Score Based Weight Categories

Students with a BMI z-score below 0 had a significantly lower eating rate (g/min) vs. students who had a BMI z-score equal to or above 0 (28.2 g/min vs. 35.9 g/min respectively, mean difference = −7.7 g/min, *p* = 0.012, 95% CI −14 g/min–−1.8 g/min; see Figure 5).

##### Objective Eating Rate among Self-Reported Slow, Intermediate, and Fast Eaters

There was a significant difference in objectively measured eating rate (g/min) between the self-reported eating rate categories [F(2, 111) = 7.104, *p* = 0.001, partial η^2^ = 0.113]. Post hoc comparisons using Bonferroni revealed that there was a statistically significant difference between slow and fast [mean difference = −13.7 g/min, 95% CI = −22.5 g/min to −4.84 g/min; *p* = 0.001)]. However, the difference between slow and intermediate and between intermediate and fast were not significant (Figure 6).

##### Categorical Agreement: Self-Reported vs. Objective Eating Rate Categories

The weighted kappa value for self-reported eating rate categories vs. objectively established eating rate categories in Study 1 was 0.31 (*p* < 0.001).

#### 3.3.2. Study 2. Repeated School Lunch Study

##### Categorical Agreement: Self-Reported vs. Objectively Established Eating Rate Categories Lunches 1 and 2

The weighted kappa value for self-reported eating rate categories vs. objectively established eating rate categories during lunch 1 was 0.36 (*p* = 0.009), while the weighted kappa value for self-reported eating rate categories vs. objectively established eating rate categories during lunch 2 was 0.30 (*p* = 0.036).

##### Categorical Agreement: Self-Reported Eating Rate Categories Lunch 1 vs. Lunch 2

The weighted kappa value for self-reported eating rate categories during lunch 1 vs. lunch 2 was 0.62 (*p* < 0.001).

##### Categorical Agreement: Objective Eating Rate Categories Lunch 1 vs. Lunch 2

The weighted kappa value for objectively established eating rate categories during lunch 1 vs. lunch 2 was 0.62 (*p* < 0.001).

##### Test–Retest Reliability of Objective Eating Rate: Lunch Meal 1 vs. Lunch Meal 2

Objectively measured eating rate during lunch 1 was significantly correlated with objectively measured eating rate during lunch 2 (Pearson’s correlation = 0.75, 95% CI: 0.59–0.85); see Figure 7.

There was a systematic change in the mean objectively measured eating rate from lunch 1 to lunch 2 (+4.4 g/min, 95% CI: 0.7–8.1 g/min) and the typical error of measurement for eating rate from lunch 1 to lunch 2 was 24.9% (95% CI: 20.4–31.9%). For more information on individual changes in objectively measured eating rate (g/min) from lunch 1 to lunch 2, see Figure 8.

#### 3.3.3. Study 3. BigO Cohort Study

##### The BigO Cohort: BMI z-Scores among Self-Reported Eating Rate (Categorical)

There was a significant difference in BMI z-scores between the three groups of self-reported eating rate in the BigO cohort [F(2, 1829) = 9.724, *p* < 0.001, partial η^2^ = 0.011]. Post hoc comparisons using Bonferroni revealed that there was a significant difference between slow and intermediate [mean difference = −0.23, 95% CI = −0.43 to, −0.03; *p* = 0.021)], as well as slow and fast [mean difference = −0.37, 95% CI = −0.57 to −0.17; *p* < 0.001)] groups. However, the difference between intermediate and fast was not significant (Figure 9).

When dividing the BigO cohort population into groups of Swedish (*n* = 748) and Greek (*n* = 1084) students, there were also significant differences in BMI z-scores between the three groups of self-reported eating rate [Swedish students: F(2, 745) = 5.955, *p* = 0.003, partial η^2^ = 0.012; Greek students: F(2, 1081) = 6.533, *p* = 0.002, partial η^2^ = 0.016]. Post hoc comparisons using Bonferroni revealed that there was a significant difference between Swedish slow and fast [mean difference = −0.36, 95% CI = −0.61 to −0.10; *p* = 0.003)] groups, Greek slow and intermediate [mean difference = −0.29, 95% CI = −0.57 to, −0.02; *p* = 0.032)] and Greek slow and fast [mean difference = −0.41, 95% CI = −0.69 to −0.14; *p* = 0.001)] groups. However, the difference between Swedish slow and intermediate, Swedish intermediate and fast, as well as Greek intermediate and fast eaters were not significant (Figure 10A,B).

When analyzing students who were treated at the obesity clinic in Athens (*n* = 304) separately, there were no significant differences between the three groups of self-reported eating rate.

## 4. Discussion

This study assessed the concurrent validity of self-reported eating rate compared to objectively measured eating rate and the test–retest reliability of such measurements in a “real-life” school lunch context. Additionally, the association between eating rate and indices of obesity (i.e., BMI z-scores and increased food mass intake during school lunch) was investigated among Swedish and Greek high-school student populations.

The finding of a “large” [45] correlation between objectively measured eating rate and short-term food intake is in line with a study in 4.5-year-old Singaporean children [15], as well as experimental studies on effects of eating rate on short-term energy intake [13]. Interestingly, when dividing students into tertiles based on their objectively measured speed of eating, students who were in the “fast” eating rate tertile ate 247 g more food vs. students in the “slow” eating rate tertile (~117% increase). This finding is in line with results obtained in the study that included Singaporean children [15]. Furthermore, students with BMI z-scores ≥ 0 had ~27% higher eating rate during school lunch vs. students with BMI z-scores < 0. Taken together, these results corroborate the suggestion of a “obesogenic” fast eating style that has previously been reported in younger children [48]. In other words, students who eat faster than their peers might be at greater risk of developing obesity, most likely due to a combination of genetic and environmental factors [49]. However, these results should be interpreted with caution, since students with increased BMI z-scores most likely have higher resting energy expenditure due to their larger bodies [50]. In other words, students with BMI z-scores > 0 might need to eat faster vs. students with BMI z-scores < 0 given the same amount of time to eat (here pre-defined by the time allocated for lunch by the school), to cover their energy needs [50,51].

We could also confirm the results observed among older populations in laboratory settings [24,25,26], that self-reported eating rate categories (slow, intermediate, and fast) could distinguish differences in objectively measured eating rate on a group level. More specifically, students who reported eating faster than others had ~56% increased objectively measured eating rate vs. students who self-reported eating slower than others, on a group level. Self-reported eating rate categories could explain ~11% of the variance in objectively measured eating rate in our study (partial η^2^ = 0.113). We are the first to confirm such findings in a “real-life” setting (i.e., the school canteen environment). However, in accordance with previous results [24,26], there was “minimal” agreement [46] between self-reported eating rate categories and objectively established eating rate categories on an individual level.

The above findings support the notion that self-reported eating rate could be used in larger-scale epidemiological studies to differentiate populations with different objective eating rates. On the other hand, self-reported eating rate should not be used on an individual level, i.e., in the context of clinical management of obesity in childhood or adolescence. It should be noted that in practice, due to monetary and time cost requirements, when it comes to monitoring of larger populations, the use of self-reported measures is often the only option. However, we argue that objective measures of eating rate should be used when eating rate is considered on an individual level.

Furthermore, when we tested the same individuals at consecutive time points (3 months apart), there was “moderate” agreement between self-reported eating rate categories during lunch 1 vs. lunch 2, as well as between objective eating rate categories during lunch 1 vs. lunch 2. These results suggest that both objective and self-reported eating rate categories are moderately stable on an individual level. Additionally, on a group level, self-reported eating rate categories could distinguish group level differences in objectively measured eating rate during both lunch 1 and lunch 2 (i.e., subjects who self-reported eating slow had lower objective eating rate vs. subjects who self-reported eating fast on a group level).

When assessing the test–retest reliability of the more granular measurement of objective eating rate (expressed as grams of food eaten/minute), there was a “large” correlation between repeated measures of objective eating rate during lunch 1 and lunch 2, indicating that objective eating rate in one lunch meal could be used to predict the rank of students eating rate during the second lunch meal with high accuracy. On the other hand, there was a systematic bias (~15%) between the two lunches (students were eating their meals 4.4 g/min faster during lunch 2 vs. lunch 1) although the same students (*n* = 50) participated in both lunch 1 and lunch 2 (i.e., within subject design) and the study setting was almost identical (i.e., with the same time duration available to eat, identical food choices, same time of the day, etc.). However, in Study 2, *n* = 37 students were eating their lunch without a food scale (i.e., the food was weighed at the food buffet before/after taking food, instead of continuously under the plate when eating food), while in Study 1, all (*n* = 50) students were eating their food on a food scale. A potential “observer effect” [52] could therefore be expected. However, food scale use at the lunch table vs. non-use at the lunch table during lunch 2 could explain 0% (R2 = 0.000) of the variation in the change in eating rate from lunch 1 to lunch 2. We therefore argue that the systematic change in eating rate must have been caused by another unmeasured environmental factor. Additionally, the typical error of measurement was similar to what has been observed for food mass intake during repeated school lunches, 24.9% in the current study vs. 26.1% for food intake in our previous study [14].

In our larger sample of students who self-reported their eating rate (i.e., the BigO cohort), we could show that students who self-reported eating faster than others had ~0.4 units higher BMI z-scores vs. students who reported eating slower than others. This finding was significant both among Swedish and Greek students and was expected based on the overall epidemiological literature [17]. However, the explanatory power of self-reported eating rate categories was low in our study and could explain ~1% of the variance in BMI z-scores. These results are in line with the results that we obtained in the single school lunch study (Study 1), in which self-reported eating rate categories could explain 1% of the variance in BMI z-scores. On the other hand, objectively established eating rate categories could explain approximately 5% of the variance in BMI z-scores, and objective eating rate expressed as grams eaten per minute could explain ~10% of the variance in BMI z-scores in Study 1. These results suggest that self-reported eating rate cannot capture the full size of the association between “real” eating rate and BMI z-scores among student populations.

Additionally, in our clinical sample of Greek students with overweight/obesity (*n* = 304), the difference in BMI z-scores among the self-reported eating rate categories did not reach statistical significance, although they had a similar tendency as our larger sample, i.e., patients with slow self-reported eating rate had lower BMI z-scores vs. patients with fast self-reported eating rate (not significant). However, the sample size was too small for such comparison since the expected effect size for eating rate categories on BMI z-scores was low. Therefore, a larger-scale study (at the scale of thousands) would be needed in a clinical student population to investigate such association. Alternatively, objective measurements of eating rate could be utilized in a clinical high-school population to increase the power and reduce the need for a larger sample size.

Our results, combined with previous studies in the area of eating rate [17,24,25,26], support the idea to include self-reported questionnaires to estimate eating rate in population-level investigations of dietary intake among students. For example, the national food agency in Sweden conducts population-level surveys about self-reported dietary intake during school lunches [53], and could include an additional question about self-reported eating rate. Such population-level information would give a more detailed understanding of the association between eating rate and BMI z-scores among the total Swedish student population, as well as enable investigations of potential regional differences in eating rates among high-schools. Similar organizations in other countries could conduct similar research without much added cost. Additionally, since objective measures are needed to assess eating rate on an individual level, large-scale studies that incorporate modern technological tools with such capability (i.e., food scales [39], off the shelf smartwatches [54] or algorithm-assisted video recordings [55]) should be considered in order to investigate the prospective association between fast eating rate with risk of long-term disease development. Such population-level data could give real-time policy advice to better manage environmental contributors to fast eating rate (i.e., school level interventions to increase time available for eating school lunch) [14].

It is important to acknowledge that the data collection in studies 1 and 3 were of an observational nature. Therefore, there is uncertainty of direction of the relationship between eating rate and BMI z-scores in our studies. Large-scale prospective studies are the logical next step and experimental studies that manipulates environmental modifiers of eating rate could also be helpful in the school lunch context. Furthermore, the students who were included in studies 1 and 2 were eating their school meals under observation (i.e., video cameras recording their meals as well as their food being measured by a kitchen scale). This addition to their normal school lunch context might have contributed to some form of “observer effect” [52,56]. Therefore, the discrepancy between objectively measured eating rate and self-reported eating rate categories might have been affected due to this setup. Covert measures of objective eating rate would be needed in the school lunch context to evaluate such effect (i.e., a study setup similar to what was used in [52]). Additionally, the questionnaire that was used to assess self-reported eating rate did not refer to a specific time point (i.e., it was framed in more general terms about their habitual eating rate), while the objective eating rate test was time and context specific (i.e., one school lunch). Also, students who had already participated in Study 1 were invited to participate in Study 2, this might have introduced self-selection bias in the results obtained from the reliability analysis. The observed bias between lunch 1 and lunch 2 (−4.4 g/min reduced eating rate) might have been related to the dates of the two lunches. Lunch 1 was conducted in the end of the semester (December 2015), while lunch 2 was conducted in the beginning of the next semester (February/March 2016). Students might have been more stressed during the end of the semester (December 2015) vs. in the beginning of the next semester (February/March 2016). The measurement of students’ body weight in close connection to the school lunch might also have modified their eating behavior (perhaps the students might have become more conscious of how they were eating). It is also important to mention that self-reported measurements of height and weight were used to calculate BMI z-scores in Study 3 and potential bias might therefore have been introduced due to this as well [57]. Furthermore, students were eating their school lunch together with other students and some form of “social facilitation” of eating might have occurred (i.e., increased food intake) since they were eating with their peers [58]. However, since students usually eat their school lunch together with their classmates, this study setup is most likely the preferred one vs. having students eat their lunch in isolation (i.e., in a laboratory-like setting). Lastly, the objective measures of eating rate in studies 1 and 2 were conducted with Swedish high-school students, meaning that the results might not be fully generalizable to the Greek student population included in Study 3 (as well as students from other countries). Additionally, IEGS is a privately owned high-school in central Stockholm and the student population might not be representative of the overall Swedish high-school student population as well as student populations in areas of lower socioeconomic status in Sweden (perhaps Uppsala NTI high-school included in Study 3).

## 5. Conclusions

Objectively measured eating rate was associated with the weight status of students, and students with fast eating rate consumed more food during school lunch vs. students with a slow eating rate. Furthermore, self-reported eating rate could distinguish student populations of different eating rates (i.e., slow eaters had lower objective eating rate vs. fast eaters). However, the agreement between self-reported eating rate categories (slow, intermediate, and fast) vs. objectively measured eating rate categories was “minimal”. Our results suggest that when objective measures of eating rate are available, those should be used instead of self-reported measures. However, in cases of studies with large sample sizes, when the behavior of the population is of interest, self-reported measures of eating rate categories could be used as a proxy of real eating rate on a group level. Furthermore, it should be emphasized that when the aim is to assess eating rate on an individual level, objectively measured eating rate should be used and self-reported eating rate categories should be avoided, regardless of the sample size. Lastly, in our larger population of Swedish and Greek high-school students, fast self-reported eating rate was associated with increased BMI z-scores vs. students who self-reported slow eating rate. The results were similar among both Swedish and Greek students.

## Figures and Tables

**Figure 1 nutrients-13-00880-f001:**
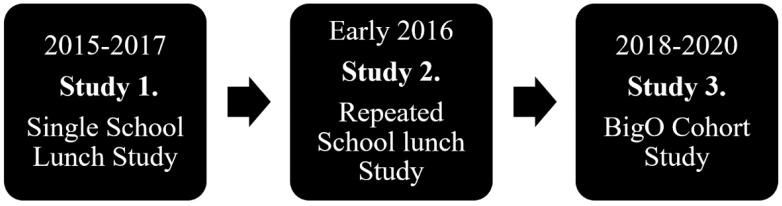
Timeline for studies 1–3.

**Figure 2 nutrients-13-00880-f002:**
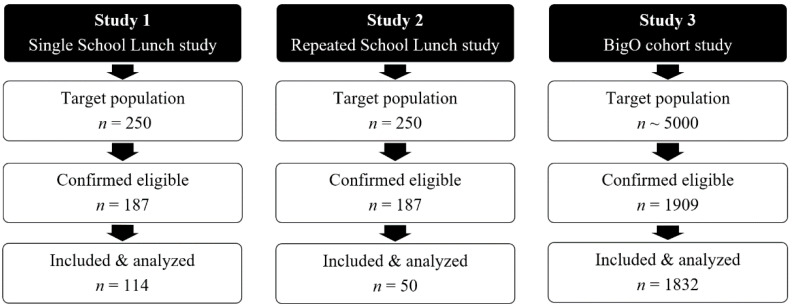
Flow diagram illustrating subjects potentially eligible, confirmed eligible, and included in this study, as well as finally analyzed for studies 1–3.

**Figure 3 nutrients-13-00880-f003:**
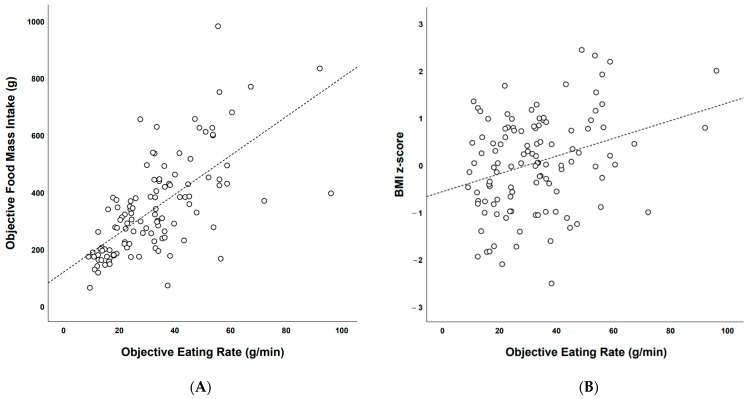
(**A**) Scatter plot illustrating the correlation between food mass intake (g) and objective eating rate (g/min) during the school lunch. (**B**) Scatter plot illustrating the correlation between BMI z-scores and objective eating rate during the school lunch (g/min).

**Figure 4 nutrients-13-00880-f004:**
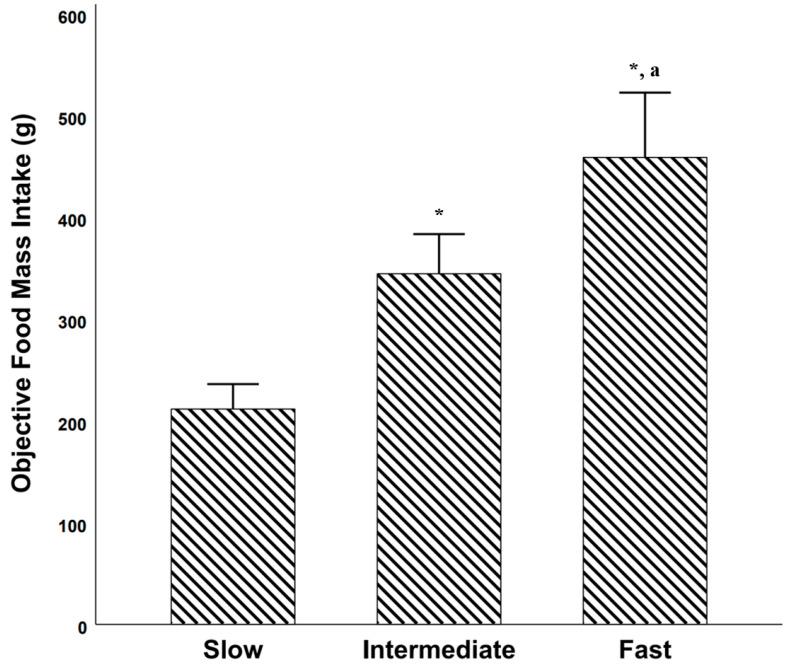
Food mass intake (g) among students who were considered as slow eaters (objective eating rate below 33 percentile), intermediate eaters (objective eating rate between percentiles 33 and 66) and fast eaters (objective eating rate above 66 percentile). * = significantly different vs. slow eaters. a = significantly different vs. intermediate eaters. Error bars represents 95% confidence intervals.

**Figure 5 nutrients-13-00880-f005:**
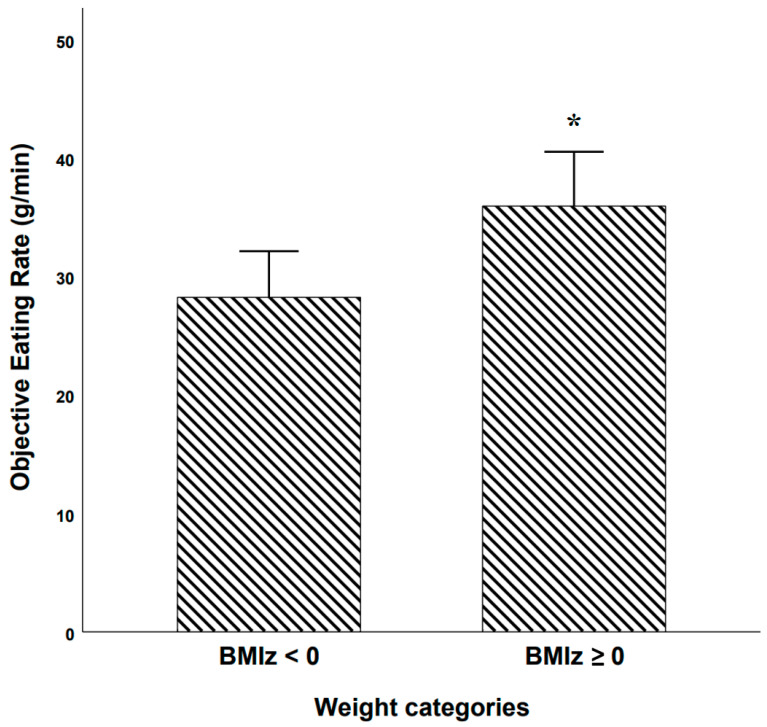
Objective eating rate among students with BMI z-score below 0 (*n* = 52) vs. students with BMI z-score equal to or above 0 (*n* = 62). * = significantly higher vs. BMI z-score < 0 group. Error bars represents 95% confidence intervals.

**Figure 6 nutrients-13-00880-f006:**
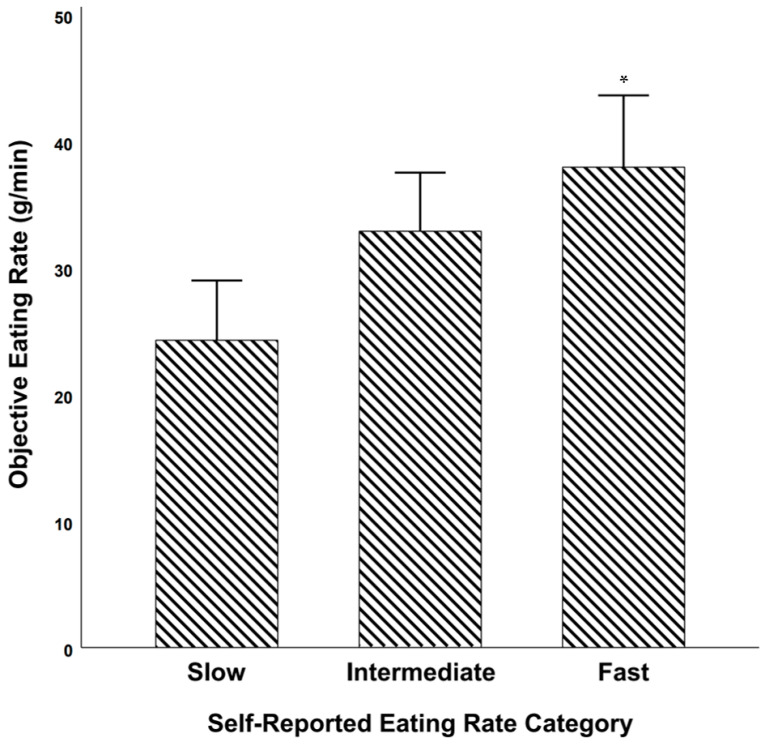
Objective eating rate during school lunch among groups of students who self-reported being slow, intermediate, or fast eaters. * = significantly higher objective eating rate vs. self-rated slow eaters. Error bars represent 95% confidence intervals.

**Figure 7 nutrients-13-00880-f007:**
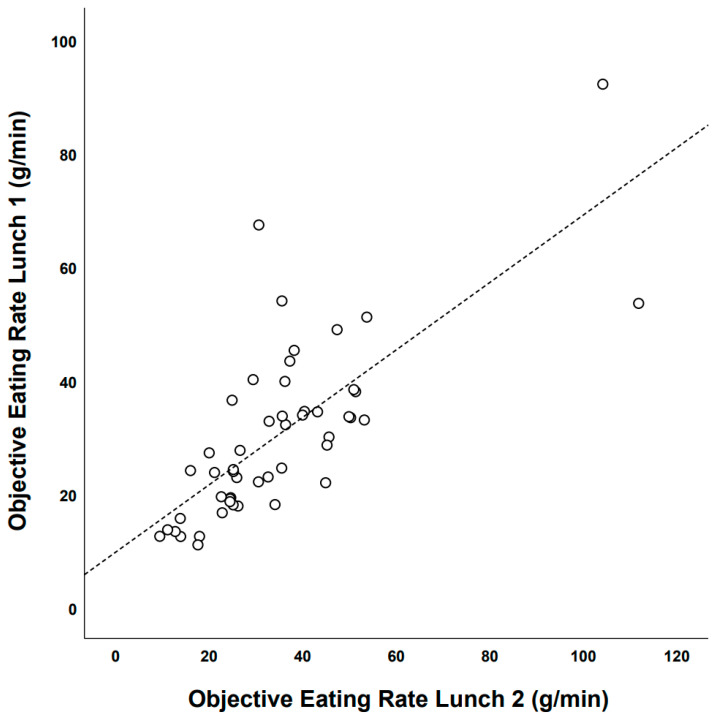
Objectively measured eating rate lunch 1 vs. objectively measured eating rate lunch 2 among IEGS students who ate repeated school lunches (Lunch 1: December 2015 and, lunch 2: February/March 2016).

**Figure 8 nutrients-13-00880-f008:**
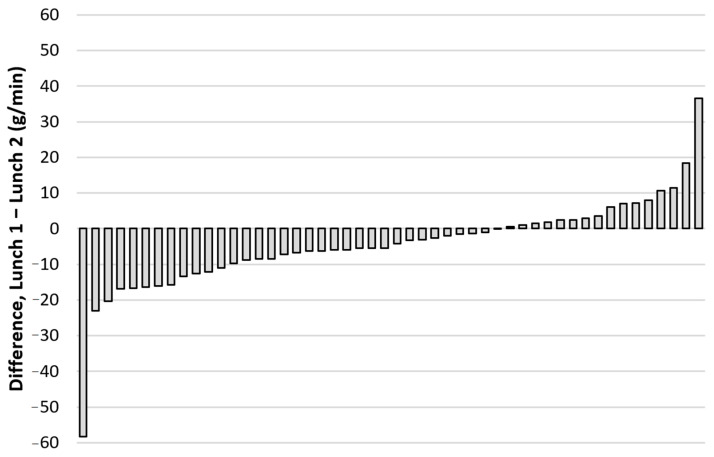
Individual changes in objectively measured eating rate (g/min) between the two repeated school lunches with identical food choices for the same subjects (*n* = 50; Lunch 1: December 2015, and lunch 2: February/March 2016).

**Figure 9 nutrients-13-00880-f009:**
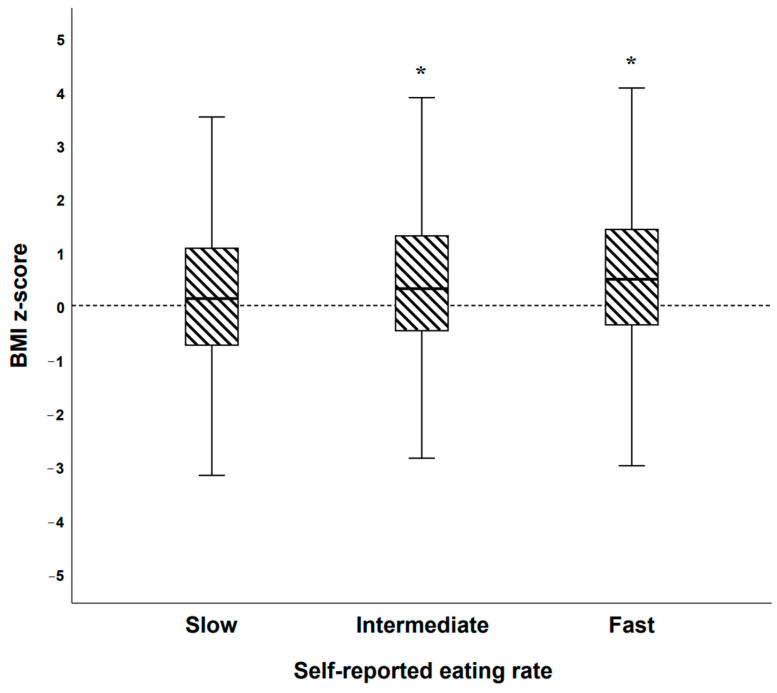
Boxplot illustrating BMI z-scores among self-reported categories of eating rate in the BigO cohort (*n* = 1832). * = significantly different vs. self-rated slow eaters.

**Figure 10 nutrients-13-00880-f010:**
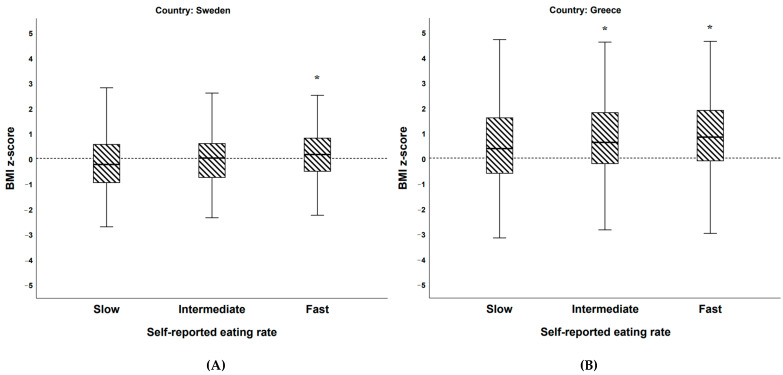
(**A**,**B**) Boxplots illustrating BMI z-scores among (**A**) self-reported eating rate categories among Swedish (**A**, *n* = 748) and (**B**) Greek (**B**, *n* = 1084) students in the BigO cohort. * = significantly different vs. self-rated slow eaters.

**Table 1 nutrients-13-00880-t001:** Descriptive statistics for the included participants in studies 1–3.

	Study 1. Single School Lunch Study	Study 2. Repeated School Lunch Study	Study 3. BigO Cohort Study
Number of subjects	114	50	1832
% females	58.8%	58.0%	51.1%
Age, y	16.5 ± 0.8	16.7 ± (0.6)	15.8 ± 0.9
BMI, kg/m^2^	21.4 ± 3.1	21.6 ± 3.4	23.1 ± 6.0
BMIz	0.04 ± 1.0	0.08 ± 1.0	0.47 ± 1.4
% with BMIz < 0	45.6%	44.0%	39.6%
% with BMIz ≥ 0 and ≤ 1.96	50.9%	52.0%	46.3%
% with BMIz > 1.96	3.5%	4.0%	14.1%
Weight, kg	62.0 ± 11.7	61.8 ± 11.8	66.6 ± 18.0
Height, cm	170.1 ± 9.8	168.7 ± 8.9	169.9 ± 10.1

BMI = body mass index, BMIz = BMI z-scores, cm = centimeters, kg = kilogram, and y = years. Data are presented as the mean ± standard deviation, if not otherwise specified.

## Data Availability

The data presented in studies 1–3 are available on request from the corresponding author.

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
