# Peer review of "Fast Eating Is Associated with Increased BMI among High-School Students"

_nutrients, 2021, doi:10.3390/nu13030880_

Round 1
Reviewer 1 Report
Overall, this is a well-written paper that provides important new information to the field of study. I also comment the authors on the clarity of the 3 studies as this can sometimes be difficult to convey. Many of my comments are meant to assist the authors in improving the readability of the paper and are not absolutely required if they are disagreed with. The methodological comments are more important to follow up with.
Lines 52-54: Perhaps the wording in this line should be plural i.e., the causes of obesity are multifactorial. Also, the second part of the sentence appears to indicate to the reader that while it is multifactorial, you can ignore that and focus on calories in = calories out, which oversimplifies the relationship. This reviewer would prefer if you just end the sentence at multifactorial.
Line 70: Not sure what SDS refers to. Is it standard deviations? If so, perhaps just using SD will be clearer to the reader.
Introduction: Authors repeatedly mention that no studies have been conducted in high-school students; however, some studies (conducted by the authors themselves) have been done in adolescents. There is likely a reason for this, nevertheless, I would suggest reducing the number of times the statement about high-school students is mentioned.
Lines 119-120: These students were from one high school and because if it’s name I wonder if it was a high school that is representative of the population or if it was a unique school. This may be important in terms of generalizability and should be noted. I see that you have done this in the discussion – thank you.
Figure 1 seems unnecessary, but authors and editors can decide to include or not.
Lines 138-140: This reviewer is surprised that participants from a pilot study were included in the sample for study 1. My Ethics board doesn’t allow this. Similarly, 2 additional students from another time period seems arbitrary. I would suggest analyzing the data without these additional students (17 in total I believe). If the outcomes are largely the same then perhaps you can make a case for including all of them, but if the results are different then I would suggest omitting these additional participants.
Line 148: You refer to the participants as school children but they are adolescents, so I would reword.
Line 183: It would be helpful for readers if you briefly explain why BMI z-scores < or > than zero were used.
Line 194: ‘in order’ is not required in this sentence and could be removed.
Line 194-196: ‘previously’ is used twice in this sentence and could be removed in one or the other location.
Line 241: For test-retest reliability, did you consider Intraclass Correlation Coefficient? I think that is a more useful statistic for this analysis as it doesn’t overestimate the reliability.
Figure 4: as a suggestion, I and others, often use a different system to show significant differences with post-hoc tests. For example, each bar in the graph would be assigned a letter and if letters are different then there is a significant difference between bars. In your case, I believe the bars would have labels a, b, c (this is not my paper, but you could look at it to see what I mean, doi:10.1017/S0007114519000370). You could also indicate what level of significance was for the comparisons in the figure legend.
Lines 393-395: It is interesting that in a group of patients being treated for obesity that there was no differences in eating rate by BMI z-score. I suppose this could be due to lack of variability in z-score, but I also wonder if there are other factors that could be driving the association you see between z-score categories and eating rate categories. Are you able to do any regression or co-variate analyses to see if there is anything else driving the relationship? I can now see that you have addressed this in the discussion, but it is quite difficult to understand without referring back to the results. I am not entirely sure what to suggest, except that perhaps you should provide more details of the regression/correlation in the results.
Line 413: Not sure what ‘this time in high school students’ means. Is there a word missing?
Line 426: This 11% explanation is quite important and I don’t recall reading it in the results. It is likely there, but perhaps it doesn’t stand out as much as it should. Can you make this more clear in the results?
Ref #31 appears incomplete.
Link for ref #39 is not available
A journal article would be a preferred ref for #43 rather than a self-published website.
Reviewer 2 Report
With this manuscript authors aims to demonstrate the association between eating fast and increased body weight. Authors have designed a three step study to evaluate: 1) the association between objective eating rate and body mass index; 2) the validity of the subjective eating rate; 3) differences in BMI according to subjective eating rate categories. In the first study took part 114 teenagers, in the second fifty participants from the first study were again evaluated; the third study comprised 748 Swedish and 1084 Greece students were enrolled. Authors found a correlation between body weight and fast eating rate (the faster the higher BMI). Although self report measures appeared weakly correlated with objective measures, fast self-reported eating rate was associated with high BMI-z scores. Considering the high prevalence of obesity, assessing possible altered eating behaviors among adolescents is an interesting and important topic. Although it is an intriguing study, I have some concerns and suggestions for authors:
- Introduction: this section is well organized and provides a good background for the research. The authors describe the health problems associated with obesity, the association between fast eating and obesity, and bring to the attention of the reader the problem on self-reported measured of dietary intake. Nevertheless authors did not take into consideration the relationship between fast eating and eating disorders. Fast eating is included among the diagnostic criteria for binge eating disorder (BED) -which in turn is frequently associated with obesity- and patients with bulimia nervosa and BED tend to fast eat during binges. Conversely, patients with anorexia nervosa tend to eat very slowly. I think this should be included in the introduction.
- Materials and methods. The methods are well described and the research can be replicated according to the explanations. I find useful BMI-Z. Nevertheless there are some questions for authors:
- Line 141-142: which was the meal you included in the dataset for study 1?
- Line 154-156: were these additional 304 participants overweight/obese?
- Line 170-172: this is an important limit that must be acknowledged. Authors are measuring the association between subjective BMI and subjective fast eating! So this is an important bias in this research.
- Line 173-175: After such a great effort with this study, I cannot understand why the authors did not consider possible exclusion criteria in their design. For example, excluding medical or psychiatric disorders or drugs that may account for fast / slow eating among participants was necessary to avoid significant bias. It would have been necessary to rule out eating disorders, which have a typically adolescent onset, that could influence the outcome of the study, among others. Authors should justify more in depth their "non-discriminatory fashion" decision.
.
- Statistical design: is appropriate
- Results: are clear. Figures and tables are clear.
- Discussion: Considering that other research has shown previously that the fast feed is associated with greater weight, the first paragraph of the discussion is excessively pompous. The authors should consider rewriting it in lighter tones.
- English language is quite good.
Reviewer 3 Report
Overview and summary of strengths/weaknesses:
This manuscript reports three studies on the associations between self-reported eating rate, objective eating rate, BMI-Z scores, and food intake. Study 1 showed a) a large correlation between eating rate and food intake; b) a moderate correlation between objective eating rate and BMI-Z scores; c) differences in food mass intake according to objective eating rate categories; d) differences in objective eating rate among BMI-Z categories; e) differences in objective eating rate among self-reported slow, intermediate, and fast eaters; and f) low agreement between self- versus objective eating rate. Study 2 was a repeat of Study 1 with a much smaller sample size and participation rate. Again, agreement between self- versus objective established eating rate category was low. There was fairly good agreement between self-reported eating rate category from lunch 1 to lunch 2 as well as for objective reported eating rate category from lunch 1 to lunch 2. There was also a large correlation between objective eating rate from lunch 1 to lunch 2. Interestingly, objectively measured eating rate decreased from lunch 1 to lunch 2. Study 3 showed differences in BMI-Z scores among self-reported eating rate category; however, the effect sizes were very small.
Strengths included the high participation rate in Studies 1 and 3 as well as the presentation of three studies in a methodologically interesting sequence. The statistics were supported by the use of clear figures.
Limitations included the low participation rate in study 2, the self-selected nature of the samples in studies 1 and more so in study 2, the use of objective measures (including portable food scales) known to the participants, and measurement of weight prior to school lunch participation.
Critique:
Study 1: Measuring student weight prior to school lunch measurements (lines 178-179) could impact eating rate (e.g., heavier students may try to eat more quickly to hide their eating). Please list as a study limitation and/or discuss.
Studies 1 and 2: It would have been helpful to measure the presence of companions or time spent talking during lunch (e.g., eating alone vs eating with others impacts energy intake, Hetherington et al., 2006, Physio Behav). Please list as a study limitation or suggestion for future research.
Studies 1 and 2: It should be noted that self-reported eating rate did not refer to a specific time point whereas the objective eating rate was specific to a single meal. Thus, self-reported eating rate could refer to general eating rate for which the specific meal was not representative. This point should be listed as a study limitation or discussed in the context of these findings (e.g., ~430).
Studies 1 and 2: Students were invited to participate in monitored school lunches (lines 136-137). Particularly in Study 2, this may have caused a self-selection bias and should be mentioned as a study limitation.
Study 2: The time period between meals was 2-3 months and there was a trend for a decrease in eating rate from lunch 1 to lunch 2. Often, there is more stress upon students as semesters/school years lengthen. Depending on the time of year, students may have purposefully eaten more quickly to allow more time to study or accomplish other tasks during their lunch breaks. Check the dates in the data to see if this might have been a factor.
Study 3 was partially conducted during the COVID-19 pandemic. Consider commenting or discussing the proportion of data collected after Jan 1, 2020. This reviewer is curious if the pandemic had any impact on eating rate or differences in eating rate according to BMI-Z score (e.g., heavier teens feeling more vulnerable and eating more quickly).
Please use the possessive of students in line 451.
